# A Review on Chikungunya Virus Epidemiology, Pathogenesis and Current Vaccine Development

**DOI:** 10.3390/v14050969

**Published:** 2022-05-05

**Authors:** Thaise Yasmine Vasconcelos de Lima Cavalcanti, Mylena Ribeiro Pereira, Sergio Oliveira de Paula, Rafael Freitas de Oliveira Franca

**Affiliations:** 1Department of Virology and Experimental Therapy, Fundação Oswaldo Cruz/Fiocruz, Recife 50740-465, PE, Brazil; thaiseyasmine@gmail.com; 2Universidade Federal Rural de Pernambuco, Recife 52171-900, PE, Brazil; myribeirop@gmail.com; 3Department of Biology, Universidade Federal de Viçosa, Viçosa 36570-900, MG, Brazil; depaula@ufv.br

**Keywords:** chikungunya, pathogenesis, virus receptor, vaccine, immunopathogenesis, epidemiology

## Abstract

Chikungunya virus (CHIKV) is a mosquito-borne alphavirus that recently re-emerged in many parts of the world causing large-scale outbreaks. CHIKV infection presents as a febrile illness known as chikungunya fever (CHIKF). Infection is self-limited and characterized mainly by severe joint pain and myalgia that can last for weeks or months; however, severe disease presentation can also occur in a minor proportion of infections. Among the atypical CHIKV manifestations that have been described, severe arthralgia and neurological complications, such as encephalitis, meningitis, and Guillain–Barré Syndrome, are now reported in many outbreaks. Moreover, death cases were also reported, placing CHIKV as a relevant public health disease. Virus evolution, globalization, and climate change may have contributed to CHIKV spread. In addition to this, the lack of preventive vaccines and approved antiviral treatments is turning CHIKV into a major global health threat. In this review, we discuss the current knowledge about CHIKV pathogenesis, with a focus on atypical disease manifestations, such as persistent arthralgia and neurologic disease presentation. We also bring an up-to-date review of the current CHIKV vaccine development. Altogether, these topics highlight some of the most recent advances in our understanding of CHIKV pathogenesis and also provide important insights into the current development and clinical trials of CHIKV potential vaccine candidates.

## 1. Introduction and Epidemiology

Chikungunya virus (CHIKV) is a mosquito-borne alphavirus. The virus was first isolated from the serum of an infected patient during a large outbreak of a debilitating arthritic disease in 1952 in Tanzania [1]. The name “chikungunya” is a descriptive term used by the local Tanzanian Makonde people to describe the disease and can be translated as “disease that bends up the joints” [1,2]. For the following 50 years (approximately) after its initial isolation, CHIKV caused only occasional outbreaks in Africa and Asia. Although CHIKV mortality rates are low, this virus imposes pronounced morbidity resulting in a substantial impact on the quality of life of infected individuals and significant economic losses, especially in developing countries. In the vast majority of individuals, CHIKV infection is characterized by an abrupt onset of fever, frequently associated with joint pain. Other symptoms are also reported, although to a minor extent, and these may include incapacitating polyarthralgia and arthritis, rash, myalgia, and headache [3]. Asymptomatic CHIKV infections do occur but are rare and estimated at about 3 to 28% of the infected individuals, varying between different epidemic outbreaks [4,5]. Acute symptomatic CHIKV disease resembles other common well-known arbovirus-induced diseases, such as dengue fever caused by the dengue virus (DENV) and the Zika virus (ZIKV) disease, which are frequently inaccurately diagnosed given the simultaneous circulation of the different viruses in the same location, making diagnosis challenging. Although the infection is usually a self-limited disease, some patients develop persistent joint pain that may last for months or years after the acute phase of disease [6,7].

CHIKV originated in Africa over 500 years ago and was subsequently introduced to Asia. The initial genetic analysis of the common CHIKV African lineage showed that the virus diverged into three genotypes, named West African (WA) and East/Central/South African (ECSA), and Asian. The ECSA can be further divided into the sublineage Indian Ocean lineage (IOL). These genotypes are now spread worldwide, with ECSA and Asian genotypes being those predominately found [8]. Interestingly these genotypes exhibited differences in their transmission cycles, while in Asia transmission appears to be maintained in an urban cycle with *Aedes* sp. mosquitoes, CHIK virus transmission in Africa was mostly related to a sylvatic cycle, primarily with *Aedes furcifer* and *Aedes africanus* mosquitoes [9]. In the urban cycle, the virus is primarily transmitted through the bite of female mosquito vectors of the genus Aedes, mostly *Aedes aegypti* and *Aedes albopictus*, vectors widely found in urban areas. However, vertical CHIKV transmission (mother-to-child transmission) was also reported in different outbreaks [10,11,12]. The virus is maintained in a rural enzootic transmission cycle, which occurs between various sylvatic *Aedes* (Stegomyia) mosquitoes and animal reservoirs, with nonhuman primates acting as the main reservoir host [13]. However, just like DENV and ZIKV, Chikungunya virus has become fully adapted to the urban cycles and no longer requires the presence of nonhuman primates and a sylvatic cycle for their maintenance. Thus, the urban transmission cycles of CHIKV, especially in densely inhabited tropical areas, usually result in large outbreaks, where a sustained low level of virus circulation is enough to maintain these viruses in the population.

It is believed that CHIKV infection has a low fatality rate, but since a large outbreak that occurred in 2005–2006 on the Indian Ocean island of Réunion (Figure 1), CHIKV infection assumptions have been made suggesting that CHIKV may have evolved to a more severe form of the disease with central nervous system (CNS) involvement and fulminant hepatitis cases being reported [14]. On the other hand, the gap in CHIKV severe disease may just be a consequence of the lack of data, with poorly described infectious mechanisms and associated pathologies from the early 21st century outbreaks. During this outbreak in 2006, CHIKV was estimated to infect a third of the population of 775,000, causing 237 deaths [15]. Given that prior to the Réunion outbreak CHIKV was reported only from sporadic cases and to a limited number of small outbreaks in several African countries and Southeast Asia, the magnitude of the Réunion Island outbreak has led to speculations that a new variant of the virus had emerged that would be either more virulent or more easily transmitted by mosquito vectors. Genetic analysis of the virus responsible for the 2006 epidemic showed that this virus originated in coastal Kenya in 2004, where CHIKV isolates obtained during this epidemic maintained a high degree of similarity and had over 99% identity at the nucleic acid level, forming a single clade within the Central/East African genotype [16]. Further analysis of genome microevolution during the 2006 Réunion outbreak identified an alanine to valine mutation at position 226 in the E1 envelope glycoprotein (E1-A226V) of CHIKV isolates. This mutation, found in over 90% of viral sequences from Réunion Island, was associated with a slight increase in transmission by *Ae. albopictus*, showing an enhanced ability of the virus to replicate in insect cells that are naturally low in cholesterol [17]. Thus, this mutation may have facilitated virus transmission, added to the fact that the virus was introduced in a region where the population was naïve (not previously exposed to CHIKV) and where mosquito population control measures were not enough to stop the virus from spreading.

Before the 2005–2006 Réunion outbreak, the virus was restricted to small African and Asian outbreaks. Of these, minor outbreaks in India were recorded in 1963 in Calcutta, followed by epidemics in Chennai, Pondicherry, and Vellore in 1964; Visakhapatnam, Rajmundry, and Kakinada in 1965; Nagpur in 1965; and Barsi in 1973 [18]. In the African continent, the first outbreaks occurred in Kenya and on the Comoros Islands (located on the southeastern coast of Africa) in 2004 [19]. During the 2006 Réunion outbreak, a large number of travelers from industrialized countries became infected, remaining infected when they returned home, which resulted in CHIKV being dispersed to other countries, and in 2007 CHIKV was identified in Italy, where a total of 205 cases of confirmed CHIKV infection were reported between 4 July and 27 September 2007 [20]. Other outbreaks were later reported in Malaysia in 2008–2009 and Bangladesh in 2008, where the CHIKV isolates were similar to the variants isolated in Cameroon, Indian Ocean islands, India, Italy, and Gabon. All these outbreaks were triggered by the emerging Indian Ocean lineage (IOL), a monophyletic lineage descendant from the ECSA phylogroup, which contains the mutation E1-A226V enhancing virus transmission by *Ae*. *albopictus* mosquitoes. Subsequently, CHIKV autochthonous cases of chikungunya fever were reported in France in 2010 [21]. Thus, CHIKV epidemiology took a dramatic turn after 2004 when this new epidemic virus strain emerged from the ECSA enzootic lineage, leading to virus dispersal among different countries.

In Asia, the first CHIKV case was reported in 1961 in Cambodia, probably caused by the Asian genotype that was circulating in the region at that time [22]. Later the virus was detected in the Philippines in 1965, then in Vietnam in 1966 and 1967, reaching Indonesia in 1972. CHIKV then spread to many territories, including Thailand, Malaysia, Sri Lanka, Singapore, and others (please refer to Figure 1 for a complete description of CHIKV transmission), causing small outbreaks or only sporadic cases. However, after 40 years, CHIKV re-emerged in Sri Lanka in 2007 in an outbreak responsible for more the 37,000 suspected cases [23]. From the perspective of the genetic epidemiology of CHIKV in the Asian continent, the ECSA strain was the most prevalent genotype found in Thailand, Malaysia, and Singapore during the 2006–2010 period, whereas from 2007 to 2014, the Asian genotype was the most prevalent strain circulating in Indonesia and the Philippines [8].

Small CHIKV outbreaks began being reported in the Americas in December 2013 when the French National Reference Centre for arboviruses confirmed autochthonous cases on Saint Martin Island (an island in the northeast Caribbean Sea); a few days later, about 50 confirmed cases were reported with additional evidence for autochthonous transmission on the island of Martinique. Then, in January 2014, autochthonous cases were also reported in Guadeloupe, Saint Barthelemy, Dominica, and the British Virgin Islands [24]. Interestingly, while in most parts of the Americas both *Aedes aegypti* and *Ae. albopictus* mosquitoes are commonly found, during the Caribbean outbreak *A aegypti* was the unique potential vector locally reported [24]. From a discrete emergence in the Americas, CHIKV turned into a major public health problem and, by the end of December 2015, nearly one million cases had been notified in the Americas, resulting in 71 deaths, and autochthonous transmission had been confirmed in more than 50 territories. Sequence analysis confirmed that the CHIKV isolated from Saint Martin belonged to the Asian lineage [24]. Subsequent analysis of CHIKV sequences from the Caribbean [25], Trinidad [26], Colombia [27], Panama [28], and Brazil [29] also confirmed the circulation of the Asian lineage. However, in 2015, in Brazil, the ECSA genotype was found in an individual who traveled to Angola and returned to Brazil where he developed symptoms; thus, for the first time during an outbreak, the ECSA genotype was reported in the Americas [29]. In Brazil, *Ae. aegypti* is predominately found in the north, north-east and central regions and is more widespread, whereas *Ae. albopictus* is more common in the cooler Southern areas of the country [30]. Thus, in 2016, the first report of naturally infected *Ae. aegypti* with the chikungunya virus genotype ECSA in the Americas was made available, supporting the hypothesis that *Ae. aegypti* mosquitoes were acting as a vector involved in CHIKV outbreaks in the northeast of Brazil [31]. This is somehow expected since, in Brazil, *Ae. Aegypti* is well known as the main vector for DENV and ZIKV and, according to the Ministry of Health, *Ae. aegypti* is the only CHIKV vector so far reported in Brazilian territory.

In 2015, in the Americas, there were 37,480 CHIKV confirmed cases reported to the Pan American Health Organization (PAHO) regional office, then, in 2016, a total of 146,914 laboratory-confirmed cases were registered. American countries reporting most cases were Brazil, Bolivia, and Colombia. In Brazil, the CHIKV outbreak was aggravated by the concomitant circulation of the other arboviruses DENV and ZIKV, where many patients experienced a dual infection. In fact, initial surveillance studies of arboviral diseases among febrile patients in Brazil identified a significant proportion of patients with serological and molecular evidence of concomitant arbovirus infection [32]. Additionally, small cohort studies identified a few patients with CHIKV severe disease experiencing a dual infection, either coinfected with ZIKV [33] or DENV [34]. It is of note that in a previous study of adults who had developed a new neurological condition following suspected ZIKV infection, at least four patients also had CNS infection with CHIKV [35]. In fact, the proportion of cases experiencing a dual infection is as high as 36%, as reported by others [36]. By the end of 2020, the Pan American Health Organization reported a total of 103,002 CHIKV cases in the Americas; these numbers were likely impaired by the SARS-CoV-2 pandemic that disorganized the entire health and surveillance systems in many countries.

To the present day, CHIKV continues to be responsible for significant outbreaks worldwide and no specific treatment or vaccines are available to prevent infection. The virus is now fully adapted to an urban transmission cycle, which represents a tremendous risk for many tropical and temperate regions. As demonstrated in Figure 1, many countries, from different parts of the world, have reported current and/or previous CHIKV transmission cases. Thus, the intensification and expansion of vector-borne diseases are likely to be a significant threat posed by climate change. In fact, although many other complicating factors (like mosquito range limits and viral evolution) exist, climate change will lead to a massive increase in exposure to Aedes-borne viruses, as predicted by several modeling studies [37,38], in which climate change will result in vector expansion toward temperate zones. This scenario is of major concern since Aedes-borne virus expansion into regions that lack previous exposure is a serious risk for public health, given the potential for explosive outbreaks when arboviruses are first introduced into naïve populations. The threat is aggravated by the cocirculation of other arboviruses that cause similar symptoms in infected individuals, resulting in novel challenges for its diagnosis and treatment. Moreover, the introduction of new strains to a susceptible population may result in an explosive number of cases, saturating the already precarious health systems of developing countries. All the above is also aggravated by the lack of proper treatment and effective control measures (i.e., vaccines and vector control). In this review, we aim to provide an update concerning the epidemiology, pathogenesis, and current status of potential novel vaccines. Although this topic has already been addressed by others, in this review we aim to bring a novel perspective of still underexplored topics on CHIKV infection; these include the current status of vaccine development, which is worth being frequently updated upon, and also the potential of neurological complications on CHIKV infected patients and the potential pathogenic mechanisms of CHIKV.

## 2. Pathogenesis

CHIKV is an alphavirus of the Togaviridae family, the virus belongs to the “new world” group of alphaviruses that includes Eastern and Western equine encephalitis, as well as the Venezuelan equine encephalitis (VEE), viruses. The virus has an approximately 12 kb positive-sense RNA genome that encodes two polyproteins, subdivided into four non-structural proteins (nsP1–4) and five structural proteins (C, E3, E2, 6K, and E1). Medically important alphaviruses that cause inflammatory musculoskeletal disease in humans with debilitating symptoms, such as arthralgia, arthritis, and myalgia are classified as “Arthritogenic viruses”; these include chikungunya virus, Ross River virus (RRV), Barmah Forest virus (BFV), O’nyong-nyong virus, the Sindbis group of viruses, and Mayaro virus (MAYV) [39,40]. CHIKV infects multiple cell types, including dendritic cells, macrophages, synovial fibroblasts, endothelial cells, and myocytes. In humans, it also infects osteoblasts, contributing to the joint pathology and erosive disease seen in chronic arthritis patients [41].

In the virion surface, heterodimers of the structural proteins E1 and E2 proteins trimerize to form the “viral spikes”, the glycoprotein E2 is responsible for receptor binding, and E1 for membrane fusion [42]. Given that CHIKV infects a wide range of cell types, cellular proteins interacting with the virus are diverse. Thus, multiple attachment factors and putative receptors for CHIKV and other alphaviruses have been documented [43]. Already known CHIKV mammalian cell receptors include prohibitin (PHB) [44], the phosphatidylserine receptor TIM-1 [45], MXRA8 [46], CD147 protein complex [47], C-type calcium-dependent lectin DC-SIGN (DC-specific intercellular adhesion molecule-3-grabbing non-integrin) [48], and, more recently, the four-and-a-half LIM domain protein 1 (FHL1) [49]. In mosquitoes, the ATP synthase β subunit was reported as a receptor for CHIKV [50]. Additionally, other phosphatidylserine binding proteins, such as Axl and TIM-4, have also been demonstrated to facilitate CHIKV infection [51]. However, a recent independent study showed that only TIM-1, unlike Axl, is functional as an entry factor for CHIKV infection [52].

Alphaviruses enter host cells by a process dependent on clathrin-mediated endocytosis [53]. More recently, micropinocytosis (which are large uncoated vesicles involved in the unspecific uptake of extracellular material) was identified as an entry pathway for CHIKV into human muscle cells. In this study, CHIKV particles were observed to colocalize with a macropinosome marker. Moreover, cell treatment with various inhibitors of endocytosis, including monodansylcadaverine (receptor-mediated endocytic inhibitor), dynasore (clathrin-mediated endocytic inhibitor), as well as filipin (caveolin-mediated endocytosis inhibitor), resulted in the minimal inhibition of CHIKV infection in vitro, whereas the treatment with a macropinocytosis inhibitor, as well as the knockdown of a protein involved in macropinosome formation, resulted in a significant reduction in viral titers [54]. In addition to this, other non-specific entry pathways were also described. It seems that the engulfment of apoptotic blebs is an important infectious mechanism. It was demonstrated that in HeLa cells, as well as primary fibroblasts, CHIKV triggered apoptosis through intrinsic and extrinsic pathways, resulting in the formation of apoptotic cell blebs (an irregular bulge in the plasma membrane that eventually blebs off). By hiding inside these apoptotic blebs, CHIKV was able to infect the neighboring cells. Thus, these apoptotic blebs act like Trojan horses, being capable of infecting macrophages and, interestingly, this process of viral replication in macrophages originating from cell blebs did not yield a proinflammatory response, constituting a mechanism by which CHIKV invades host cells and escapes the host response [55].

After infection, the incubation period ranges from three to seven days. Laboratory diagnosis is achieved by IgM-capture ELISA and/or reverse-transcription polymerase chain reaction (RT-PCR) assays from blood samples. During the acute phase, the most commonly reported symptoms are high fever, rigors, headache, photophobia, and a petechial rash or maculopapular rash. In comparison to the other well-studied arboviruses DENV and ZIKV, CHIKV infected patients experience a much higher viremia, in which the peak of viremia coincides with the fever onset [56,57]. In addition, the viral load seems to be a good predictor of disease severity, since hospitalized cases had higher viremia than those who did not require hospitalization [56]. It is roughly estimated that 30 to 40% of the infected individuals experience some long-term sequelae; these include persistent arthralgia and/or arthritis, in which severe pain is present in about 37% of individuals with persistent arthralgia [58]. Factors associated with persistence of arthralgia in CHIKV infected patients have not been fully explored, and, from the few available studies, patients aged >40 years, of female sex, and with higher levels of CXCL8 detected during the acute phase of the disease were shown to be associated with CHIK persistent arthralgia [59,60,61]. It was shown that human synovial fibroblasts were susceptible and permissive to CHIKV ex vivo experimental infection, and single-cell analysis evidenced a strong upregulation of arthritis-associated genes RANTES/CCL5 and IL-8 from infected cells [62]. In macaques, long-term CHIKV viral RNA expression was associated with mononuclear cell infiltration of the synovial tissue. In this model, the CHIKV infection chronic phase was characterized by the persistence of activated macrophages and presence of viral RNA and viral antigens for up to 2 months in lymphoid organs and liver. Additionally, at high doses of CHIKV, primarily inoculated, features of arthritis, meningoencephalitis, and death of infected animals was observed [63]. Interestingly, the inability of aged rhesus macaques to clear CHIKV experimental infection was found to be correlated with reduced innate and adaptive immune responses compared to adult animals. More precisely, the plasma from aged CHIKV-infected rhesus macaques have lower levels of bioactive type-I IFN (reflected in the diminished activation of type-I IFN stimulated genes Mx1 and ISG56), in addition to reduced anti-CHIKV IgG titers and impaired T cells, as well as B cell responses [64]. In mice, impaired type-I IFN response leads to mild disease, where, after a burst of viral replication in the liver, the virus targets muscle, joint, and skin fibroblasts; a cell and tissue tropism similar to that observed in biopsy samples of CHIKV-infected humans. In a study of neonatal mice, as well as of the adult mouse with a totally abrogated type-I IFN signaling (IFN-α/βR^−/−^), CHIKV-associated disease was particularly severe, and this severity correlates with higher viral loads and dissemination to the CNS [65]. Clearly, altogether these data demonstrate that CHIKV disease severity is critically dependent on at least two host factors: age and functionality of type-I IFN signaling.

Although advances have been achieved regarding a better understanding of CHIK infection, a precise understanding of the immunopathogenic mechanisms of CHIKV induced arthralgia remains unclear. During the acute phase, CHIKV viremia can reach a maximum of 10^6^ in adults [66] to 10^9^ RNA copies/mL in children [67], showing a direct correlation with plasma levels of certain cytokines. Thus, as for other infectious diseases, viral load might have a direct influence on the expression levels of proinflammatory cytokines and chemokines, which in turn might influence the presentation of different clinical symptoms. In another study, CHIKV viral load ranged from 2.91 × 10^4^ to 8.55 × 10^8^ RNA copies/mL, mainly observed when blood was collected within the first 3 days after the symptoms appeared [68]. These variations among independent studies may be a consequence of the different protocols applied, in which correct assumptions between viral load and disease severity may not be possible across different outbreaks. Although pathogenesis and virulence variation among CHIKV lineages and sublineages have not been fully investigated, there are a few reports in the literature. In one study, performed on suckling mice, both strains (Asian and East/Central/South African) replicated similarly; however, the Asian strain generated higher mortality and upregulation of proapoptotic genes, whereas the East/Central/South African strain caused lower mortality and higher upregulation of antiviral and antiapoptotic genes [69]. Another study demonstrated that the West African (WA) strain produced significantly higher viremia than ECSA, Indian Ocean lineage (IOL), and Asian/American strains in AG129 mice at two days post-infection [70]. In rhesus macaques, comparisons between the WA and ECSA strains found no pathophysiological differences, although the WA strain induced a modestly nonsignificant higher viremia [71]. Based on all the above we may assume that distinct CHIKV lineages vary in virulence, at least when assessed in animal experimental models.

Intriguingly, a few studies showed the presence of IgM against chikungunya virus for an extended period after infection [72,73]. Chopra et al. (2010) reported high levels of CHIKV IgM in a cohort of Indian patients with post-CHIKV rheumatoid arthritis; in this same study, a large percentage of the patients were also IgM+ for a prolonged period of 30 to 180 days post-infection [74]. In a cohort of 57 patients with confirmed infection, CHIKV-IgM was detectable in 7/57 (12.3%) patients after 28.3 months of infection [75]. Based on these findings, we can suggest that prolonged detection of CHIKV specific IgM is linked to virus persistence and although it has not been directly proven, some data from animal experimental models support this hypothesis, as discussed next.

It has been found that CHIKV chronic disease manifestation, resulting in persistent arthralgia disease with arthritis-like symptoms, is more likely to occur in elderly patients (>60 years) with much higher viral loads (up to 10^10^ viruses per milliliter of blood) during the acute phase of the disease; additionally, the IFN-α antiviral immune response and IL-12 levels persisted for months in the chronic group. Higher viral loads in the acute phase were accompanied by higher production of proinflammatory cytokines IFN-α and IL-6, and, as the disease progressed to the chronic phase, IL-17 became detectable. However, persistent arthralgia was associated with higher levels of IL-6 and GM-CSF, whereas patients who fully recovered presented higher levels of eotaxin and HGF [76]. CHIKV chronic disease was also associated with the persistence of CHIKV antigens in synovial macrophages, which was accompanied by a polarized local inflammatory response [77]. More recently, CHIKV RNA and antigens were detected up to 90 days post-infection in the spleen, lymph nodes, liver, and muscle tissue of experimentally infected macaques [63]. In Rag-1^−/−^ experimentally infected mice (which lack T and B cells) CHIKV RNA persisted in joint-associated tissues up to at least 16 weeks post-inoculation. The persistence of viral RNA in joint-associated tissues was associated with histopathological evidence of arthritis, synovitis, and tendonitis [78]. Hawman et al. (2016) demonstrated that virus persistence is also linked to a specific CHIKV pathogenic strain. The amino acid at position 82 of the CHIKV E2 glycoprotein, which varies between the strains 181/25 (attenuated) and AF15561 (pathogenic), results in viral persistence in joint tissues of animals infected with the AF15561, while the strain 181/25 is rapidly cleared. These findings suggest that pathogenic CHIKV strains evade E2 domain-B-neutralizing antibodies to establish persistence [79]. Based on all the above we can then add, virus feature variables that could result in a more aggressive infection phenotype to the host factors disease severity, of which viral persistence and viral load seem to be key players. Although these findings from animal models helped to elucidate some aspects of the molecular mechanisms of CHIKV chronic disease pathogenesis, including immune evasion mechanisms, in humans, data about CHIKV virus persistence are not clear. Thus, although the CHIKV-induced prolonged inflammation may be related to virus persistence, this has not yet been fully explored. On the other hand, assumptions can be extrapolated from the closest alphavirus, Ross River virus (RRV). From some in vitro studies, it was observed that macrophages infected with RRV were capable of producing virions for over 170 days [80,81]. Thus, the development of chronic arthritis in alphavirus infections may result from an inability of the induced immune response to completely clear the virus.

In addition to the persistence mechanism, it was demonstrated that the engulfment of CHIKV-apoptotic cell blebs (apoptotic bodies derived from infected cells) promotes the infection of neighboring cells and macrophages in a noninflammatory (or dormant) manner [55]. In another study, whole blood purified monocytes were experimentally infected with CHIKV, showing sustained virus growth, and since infected monocyte/macrophage cells have been detected in the synovial tissues of chronically CHIKV-infected patients, these cells may behave as the vehicles for virus dissemination [82]. Clearly, this process contributes to a robust and uncontrolled adaptive immune response, leading to autoimmunity against self-antigens present in apoptotic bodies. Additionally, infected macrophages may behave as Trojan horses, helping to disseminate the virus to particular sites, and some of these sites could become “virus reservoirs”, supporting persistent viral replication.

As for other viral infections, cytokines and chemokines are key players in CHIKV immunopathology. During the early acute phase, serum proinflammatory cytokines IFN-α, IFN-β, IFN-γ, CXCL10/IP-10, and IL-1β show a strong upregulation. IFN-α is detected early during infection (e.g., on the first day), and its concentration correlates with viral load, which is significantly higher in elderly patients [83]. The elevation of MCP-1, IL-6, IL-8, MIP-1α, and MIP-1β was most prominent in the chronic phase [84]. Among the initial analysis, IL-1β, IL-6, and RANTES were found to be specific biomarkers associated with severe CHIKV disease in patients from the 2007 Singapore CHIKV outbreak [85]. In another study, CXCL9/MIG, CXCL10/IP-10 levels, and high concentrations of IgG were found to be associated with severe symptoms manifesting in CHIKV patients [86]. TNF-α and IFN-γ-secreting NK-like T cells were high in patients presenting CHIKV persistent arthralgia [87]. In addition to this, a systematic meta-analysis of immune signatures in patients with acute chikungunya infection showed an association between increased proinflammatory cytokines and arthralgia [88]. However, even though cytokines and chemokines profiles play an important role in viral immunopathology, one limitation of these studies is the fact that they were based on small cohorts and the disease classification did not include patients who developed neurologic or hemorrhagic complications.

Genetic host factors also influence the progression of the infectious disease and its clinical outcome. It has been shown that Toll-like receptor (TLR) polymorphisms may influence human CHIKV-susceptibility and disease progression. More precisely, certain genotypes of TLR-7 (rs3853839) and TLR-8 (rs376487) were significantly associated with CHIKV susceptibility [89]. In a previous study, it was demonstrated that TLR3 polymorphisms rs3775290 and rs6552950 correlated to chikungunya susceptibility and disease severity, respectively [90]. Chikungunya disease was also associated with polymorphism in DC-SIGN (a c-type lectin receptor present on the surface of macrophages and dendritic cells) and TLR3 genes [91]. Taken together, these results substantiate a role for TLRs in the control of CHIKV replication, immunity, and pathology. This can be explained by the fact that CHIKV triggers a strong local type-I IFN response, and the magnitude of this response is crucial to limiting virus replication, thus an altered type-I IFN response may result in enhanced virus replication and persistence. One reasonable explanation is that, just as recently reported for SARS-CoV-2, inborn errors of immunity (IEI) are genetic determinants of severe CHIKV disease. As reviewed by Zhang et al., 2022 inborn errors of the type-I IFN response, including Toll-like receptors (TLR3, TLR7, and TLR8) as well as its downstream signaling molecules (i.e., interferon regulatory factors (IRF)), are implicated in disease exacerbation in COVID-19 patients [92]. In fact, rare IEIs have been related to disease aggravation in several different types of viral, bacterial, fungal, and parasitic infections. In a simplified way, individuals with a “not optimal” type-I IFN response, whatever the underlying determinants, may be unable to prevent virus spread to other organs during the first few days of infection, resulting in virus persistence and chronic inflammation. However, at present, a major challenge is to better understand specific IEIs and their role in Chikungunya disease. To achieve this, large cohort studies are necessary, and given the short life of the most recent outbreaks, the recruitment of patients may not be feasible. Based on the reduced number of studies focused on CHIKV neuropathogenesis some important questions remain; for example, what is the percentage of neurological complications in CHIKV-infected patients?

Even though CHIKV is not considered a true neurotropic virus (able to invade the neural tissue and replicate in the neurons), sporadic cases of neurological manifestations have been reported from different outbreaks. Clearly, the lack of a more detailed understanding of CHIKV neuropathogenesis may be a direct reflection of the occurrence of the minor outbreaks, limited to well-defined territories and small populations, that have been documented so far. In other words, since the neurological complications of CHIKV infection are rare, small outbreaks often result in a low number of neurological cases that are commonly missed by health professionals. Thus, given that CHIKV neurological complications are infrequent, this aspect of CHIKV infection started to be reported only recently. In a previous study, CHIKV neurological complications were found to be diverse and the most commonly reported neurological complications included encephalitis, myelopathy, peripheral neuropathy, myeloneuropathy, and myopathy [93]. Interestingly, in the La Réunion 2005–2006 outbreak, of the 33 patients admitted to the intensive care unit with confirmed acute chikungunya virus infection, 14 were diagnosed with encephalopathy. Laboratory analysis found the presence of CHIKV specific IgM antibodies in the cerebrospinal fluid (CSF) samples from the vast majority of these patients and also positive virus detection in the CNS from at least two cases [94]. A prospective investigation of suspected Chikungunya cases in hospitalized patients from two large cities (Ahmedabad and Pune) in India, in 2006, registered 99 cases with neurological manifestations, neurological complications included encephalitis (57 patients), encephalopathy (42), and myelopathy (14), or myeloneuropathy (12 patients) [95]. More recently, from a large cohort study from Brazil, CHIKV infection was more often associated with a central nervous system disease (CNS), mainly myelitis, while ZIKV infection (a highly neurotropic arbovirus) was more commonly associated with peripheral nervous system (PNS) disease, particularly Guillain–Barré syndrome. Viral RNA detection in CSF samples of patients presenting a neurological disease reinforced the findings that CHIKV disease is not so rarely associated with CNS infection [96]. A major challenge to correctly diagnose a CHIKV neurological case lies in the fact that often CHIKV cocirculates with other arboviruses, and, clearly, one of the substantial limitations to better studying CHIKV neuropathogenesis resides in the fact that large cohort studies are necessary to capture a sufficient number of patients presenting neurological symptoms. This is also complicated by the fact that CHIKV outbreaks are usually short-lived, often showing seasonality, which does not allow adequate time to conduct new studies.

Explorations of the pathogenic mechanism of CHIKV infection in the brain are scarce. It has been shown that SH-SY5Y cells, a human neuroblastoma cell line widely applied to model neuroinflammation in vitro, are susceptible to CHIKV infection. Following infection with a CHIKV virus belonging to east/central/south African genotypes isolated from the 2006 Indian epidemic, these cells showed features of apoptosis, such as caspase-3 activation, cleavage of PARP, and cytochrome-c release, suggesting the implication of virus-induced apoptosis in disease pathogenesis [97]. In young mice, CHIKV viral antigens and RNA were present in the cortex of experimentally infected animals at two days post-infection, resulting in an expressive host proteome modification in the brain [98]. In another study, CHIKV infection resulted in the upregulation of several interferon-regulated genes in mouse brain tissues and a human neuronal cell line IMR-32 [99]. In BALB/c mice, CHIKV virus was found to replicate in the brain with a peak titer of 10^4^ at 6 days post-infection, infecting, preferentially, neuronal cells of the cerebellum [100]. Interestingly, CHIKV infection does not result in altered blood–brain barrier (BBB) permeability. It seems that CHIKV targets primarily the choroid plexus epithelial cells, which are highly susceptible to infection via the apical route and, to a lesser extent, via the basal route, suggesting that CHIKV accesses the CNS through the choroid plexuses. The virus exhibits a marked tropism for the meninges, whereas it does not infect the brain microvessels and parenchyma and does not induce tissue alteration at the brain parenchyma level. Thus, in contrast to other well-known encephalitogenic alphaviruses, CHIKV infection is associated with reversible CNS symptoms, as a result of its inability to infect the brain parenchyma and neurons [65]. Clearly, these studies are valuable and shed some light on neurological CHIKV pathogenesis; however, they run into the lack of a more suitable animal model for CHIKV experimental infection, since they were based on neonatal and or type-I deficient mouse infection models.

## 3. Vaccines and Treatment

Although CHIKV is now endemic in many regions, at the moment no vaccines are available, and the treatment of CHIKV infected patients is mainly based on the use of anti-inflammatory drugs for symptomatic relief. Importantly, a CHIKV vaccine needs to be economical, easy to handle, and easy to transport, thus enabling developing countries to be able to afford a mass vaccination campaign. At present, there are several preclinical vaccines in development and a limited number of clinical trials are currently being carried out across the globe. Since CHIKV persistent arthralgia disease shares some clinical similarities with rheumatoid arthritis (RA), some widely employed disease-modifying antirheumatic drugs (DMARDs), such as methotrexate, sulfasalazine, leflunomide and hydroxychloroquine, have been utilized but with limited efficacy [101]. Many different antivirals have been investigated as potential therapeutics against CHIKV. Different approaches were employed by several research groups to identify novel potential CHIKV antiviral drugs and drug candidates that have been repurposed to inhibit CHIKV replication; however, none of them are currently under clinical evaluation. An up-to-date and complete review of these findings is presented by Hucke and Burget (2020) [102] and Battisti et al. (2021) [103].

CHIKV preclinical candidate vaccines under preclinical development are diverse; these include a whole-virus inactivated vaccine [104], a VEE/CHIKV chimeric vaccine [105], a recombinant adenovirus vectored vaccine [106], a DNA-based CHIKV vaccine [107], a virus-like particle (VLP) vaccine [108], and a live-attenuated vaccine that has a stronger and longer-term immune response [109]. More recently, mRNA-based vaccines became available due to their emergency use authorization during the SARS-CoV-2 pandemic; thus, as for CHIKV, pharmaceutical companies are now developing immunization strategies based on mRNA vaccines and some clinical trials are under evaluation. Although vaccination strategies against CHIKV are diverse, currently only a few vaccine candidates are under clinical trial evaluation (please refer to Table 1 for a complete list of vaccines currently under clinical trial evaluation). Since these vaccines are in a late stage of development, we will focus on discussing their preliminary results, when available, and the potential for large-scale use in the general population.

Recently, Kose et al. (2019) reported the results of a preclinical evaluation of a lipid nanoparticle (LNP)-encapsulated mRNA, encoding the light and heavy chains of a human monoclonal antibody (named CHKV-24 IgG) and targeting the CHIKV E2 glycoprotein, which elicited high levels of biologically active CHIKV-neutralizing antibodies when administered into mice or cynomolgus macaques [118]. The authors employed a strategy based on antibody therapy from previously known neutralizing antibodies, encoding their sequences into mRNA molecules that were delivered by infusion with very promising results. This study concluded a now phase 1 clinical trial, where the safety and pharmacology of mRNA-1944 (lipid nanoparticle-encapsulated mRNA, encoding the heavy and light chains of the CHIKV monoclonal-neutralizing antibody CHKV-24) in humans was evaluated; results showed both detectable in vivo expression and detectable ex vivo neutralizing activity of this formulation, representing a treatment option for CHIKV infection [117]. This strategy is supported by the recent large-scale testing of mRNA vaccines in phase 3 clinical trials that demonstrated protective efficacy against symptomatic SARS-CoV-2 infection, leading to the licensing of SARS-CoV-2 mRNA based vaccines in many countries. However, these data have some limitations since the duration of the neutralizing antibody response was assessed only up to 48 h after mRNA-1944 intravenous infusion; thus, longer-term studies are needed. Clearly, the use of this strategy is limited to therapeutic use only; therefore, we can speculate that the mRNA-1944 vaccine will not be available to low and middle-income countries, as was observed for SARS-CoV-2 mRNA vaccines. Another point that deserves discussion is that the mRNA-1944 vaccine requires intravenous administration, limiting its use to hospitals or specialized health care centers.

An early developed CHIKV vaccine, named TSI-GSD-218, was also evaluated in a clinical trial study. This is a live attenuated vaccine obtained from a serially passaged, plaque-purified live CHIKV strain derived from a serum isolate (CHIK strain 15561), obtained from an infected patient during the 1962 outbreak of CHIK disease in Thailand. Fifty-nine volunteers were immunized one time subcutaneously with TSI-GSD-218 and fourteen were immunized with a placebo formulation. Except for transient arthralgia in five vaccine recipients, the number and severity of local and systemic reactions and abnormal laboratory tests after immunization were similar in TSI-GSD-218 and placebo recipients. Induction of neutralizing antibodies was observed from a high percentage (98%) of TSI-GSD-218 vaccinees at 28 days after immunization, and 85% of these remained seropositive at one year after immunization [115]. Despite the promising results, this vaccine never advanced to a phase III clinical trial, and, given that this vaccine was developed in the year 2000, we may consider it now abandoned and it will not be available on the market.

A live attenuated vaccine based on the infectious clone of the La Réunion strain, CHIKVLR2006-OPY1, was developed [110]. This virus strain was genetically modified in the gene encoding the nonstructural replicase complex protein nsP3, which resulted in the attenuation of the virus in vivo; this vaccine was named VLA1553 and was designed by Valneva, a French pharmaceutical company. The immunization strategy was designed to protect against all circulating CHIKV genotypes and even, eventually, against other alphaviruses that are closely related to CHIKV [111,119]. In a phase I clinical trial study of VLA1553, the factors evaluated and determined were the safety and immunogenicity of the three escalating doses in adults, including revaccination after 6 or 12 months of the last dose administered. Phase I results of this study were promising, as the vaccination induced high titers of neutralizing antibodies after a single dose. The achieved seroconversion rate of 100% was maintained over one year after dosing at months 6 and 12. Vaccinees were protected from vaccine-induced viremia, which indicates its efficacy. By providing protection only after a single dose, this vaccine is promising, especially in regions that have outbreaks of CHIKV infection. Reported adverse events were predominantly headache, fever, and fatigue, followed by muscle and joint pain after immunization; however, these symptoms are commonly reported after vaccination with other vaccines in the general population and are not a major concern [120]. This vaccine is now under a phase III clinical trial in Brazil and results concerning its efficacy should be available soon.

A phase III clinical trial of a virus-like particle (VLP) CHIKV based vaccine is currently recruiting participants. This vaccine, named PXVX0317, was developed by PaxVax, a USA vaccine manufacturing company that worked in collaboration with the National Institute of Health and the Department of Defense, USA. This is a multi-center, randomized, double blind, placebo controlled phase III clinical trial study to evaluate the safety and immunogenicity of PXVX0317 in healthy adult and adolescent subjects. The primary outcome is to assess the induction of anti-Chikungunya virus (CHIKV) neutralizing antibody geometric mean titers by different formulations and schedules. Immunization strategy was designed in a two dose regimen administered in a 30 day (approximately) interval. Some adverse effects reported included nausea, injection site pain, fatigue, malaise, myalgia, arthralgia, and headache, but no serious adverse effects were reported. These effects were more frequently reported from groups who received a higher dose antigen of 20 micrograms of aluminum hydroxide gel (Alhydrogel) adjuvanted CHIKV VLP PXVX0317 on days 1 and 29, followed by 400 micrograms of aluminum hydroxide gel (Alhydrogel) adjuvanted CHIKV VLP PXVX0317 on day 547 (Clinical Trials.gov, Identifier: NCT03483961).

Results from a phase I clinical trial of the candidate recombinant chimpanzee adenovirus-vectored vaccine ChAdOx1-Chik were recently made available. ChAdOx1-Chik contains the full-length polyprotein of CHIKV, including capsid, 6k, and envelope E3, E2, and E1. Vaccine construction was based on the analysis of the full-length structural polyprotein sequences for CHIKV from all lineages, which were collected from the NCBI protein database, sequences were aligned, and the structural cassette CHIKV sequence derived from various CHIKV lineages was codon-optimized and rescued in vitro to generate the recombinant chimpanzee adenovirus-vectored vaccine [106]. Clinical trial participants received a single intramuscular injection of ChAdOx1-Chik at one of the three preestablished dosages (5 × 10^9^, 2.5 × 10^10^ and 5 × 10^10^) and were followed up for 6 months. ChAdOx1-Chik was safe at all doses tested with no serious adverse reactions documented. Adverse events were self-limited with mild or moderate intensity. Broadly neutralizing antibodies against the four CHIKV lineages were found in all participants as early as 2 weeks after vaccination [116]. An advantage of this strategy resides in the fact that adenoviruses represent the most widely used viral-vectored platform for vaccine design, showing great potential for use in the general population. Moreover, novel adenoviral vectors including nonhuman adenoviral vectors have emerged to be the further improved vectors for vaccine design and are now being increasingly explored for vaccine development. However, given the wide use of the SARS-CoV-2 vaccine ChAdOx1-S, significant adverse effects started to be reported, these include thrombocytopenia and major arterial and venous thrombosis [121] that were associated with a rare development of immune thrombotic thrombocytopenia mediated by platelet-activating antibodies against platelet factor 4 (PF4) [122], raising concerns about the safety of adenovirus vectored vaccines.

A recombinant live Schwarz strain measles-vectored CHIKV vaccine, which was designed to express CHIKV virus structural proteins, is currently under clinical evaluation. This strategy was based on the insertion of the entire subgenomic open reading frame (ORF) encoding the CHIKV structural genes (C-E3-E2- 6K-E1); CHIKV structural genes were inserted in the intergenomic region of phosphoprotein (P) and matrix protein (M) as additional transcription units into the infectious cDNA of the Schwarz measles vaccine (MV). Measles vaccination has the advantage of being safely used for more than 40 years in over 1 billion children, representing a powerful vaccine platform that has been extensively evaluated in the general population [123]. Moreover, measles vaccine strains are genetically stable and, given it is a negative-stranded RNA genome, reversion to pathogenicity or integration into the host cell genome is virtually impossible and has never been documented [124]. In preclinical trials, the MV was showed to be an effective and safe carrier for CHIKV antigens and may enhance pre-existing measles immunity [125]. In phase II, a double-blind, randomized, placebo-controlled clinical trial, the vaccine candidate showed great promise in preventing Chikungunya infection; this is attributed to its safety and immunogenicity in healthy adults. Vaccine schedule consisted of a lyophilized MV-CHIKV-202 formulation administered in two different concentrations as follows: 5 × 10^4^ and 5 × 10^5^ TCID_50_ per dose in an injectable volume of 0.3 mL water, MV-CHIKV was administered in three intramuscular injections on days 0, 28, and 196. A control group, defined by the administration of a live virus formulation containing the attenuated Schwarz measles virus strain (Priorix), was also evaluated. Study participants were randomly assigned to receive Priorix or the MV-CHIKV vaccine candidate. Preliminary data demonstrated that the high dose groups (5 × 10^5^ TCID_50_) presented significantly higher concentrations of neutralizing antibodies than lower dose regimen (5 × 10^4^ TCID_50_) immunization, although CHIKV neutralizing antibodies were detected in all MV-CHIK immunized groups, either after one or two immunizations. Seroconversion rate at 28 days after the second immunization ranged from 50% to 96% [112].

Another vaccine which is also showing promise in its clinical trial phases is the BBV87 vaccine, formulated with an inactivated whole virus, based on the east/central/south African genotype strain. This type of strategy uses whole virus nonreplicant particles that cannot convert to their virulent form. The inactivation of CHIKV can be performed by means of some components, such as: 1,5 iodonapil azide [126], ultraviolet (UV) [127], formalin, and β-propiolactone [104]. In the case of the BBV87 vaccine, the method of viral inactivation is formalin, which is commonly used for the production of inactivated viral vaccines and has also been evaluated in preclinical trials with CHIKV. These assays revealed that viral inactivation with formalin induced high titers of neutralizing antibodies and presented immunogenic potential to neutralize the viral infection, making it a promising method for the development of new vaccines against CHIKV [128]. In a phase I study of the BBV87 vaccine carried out in India, the vaccine was found to have the potential to generate a satisfactory immune response against CHIKV. The study was registered in 2017 by Bharat Biotech International Ltd. in the Clinical Trial Registry—India. Based on these results, this study has advanced and is in phase II and III clinical trials in Costa Rica. The study is recruiting healthy individuals aged between 12 and 65 years in Panama and Costa Rica and has a strategy of administering two doses of the vaccine. Phase II, blinded observer, is divided into parts A and B and will use the BBV87 vaccine (20 µg/40 µg), formulated with 0.25 mg of aluminum hydroxide, to select the dose. Phase III, represented by part C, will use a placebo comparator, saline solution, which will confirm the selected dose and analyze the immunogenicity. The geometric mean of neutralizing antibody titers and seroconversion rates will be measured 28 days after the first dose of BBV87/placebo and 28 days after the second dose. At 6 months after the second dose, only parts B and C will be analyzed (Clinical Trials Registry—India (CTRI), Number CTRI/2017/02/007755).

Overall, CHIKV vaccine development aims at the induction of neutralizing antibodies. Although antibodies are crucial for protection against many infectious diseases, for some infectious diseases suboptimal levels of antigen-specific antibodies or non-neutralizing antigen-specific antibodies may enhance infectivity and disease severity through a phenomenon known as antibody-dependent enhancement (ADE). This phenomenon has been observed in vitro for the closely related alphavirus RRV [80] and, although CHIKV ADE has not been reported to date, care must be taken.

## 4. Concluding Remarks

As CHIKV is rapidly spreading through many countries, vaccination of the susceptible population is the most powerful method to control infection. The present review highlights important considerations concerning aspects of pathogenesis, immunology, and clinical manifestations resulting from CHIKV infection. A better understanding of the mechanisms of CHIKV disease, its different clinical manifestations, and the constant interest of the scientific community in the search for a method that prevents the infection resulted in significant advances in the development of a safe and effective vaccine. In this way, in this review we emphasized a discussion of the more recent updates of the more advanced vaccine clinical trials, some of them moving to their final phase; these trials are now showing encouraging results and we may expect to have a CHIKV vaccine available to the general population very soon. As CHIKV is rapidly spreading through many countries, vaccination of the susceptible population is a powerful method to control infection. On the premise of making vaccines available to middle- and low-income countries, usually the most CHIKV affected countries so far, an ideal vaccine should be thermo stable and easy to produce, transport, and store. In addition, their administration should be convenient and long-term protection is highly desirable. Regarding pathogenesis, crucial questions to be addressed still remain; for example, the role of the immune system in the development of chronic and or neurologic diseases has yet to be deciphered. We have recently started to see an increase in neurological associated arboviruses diseases; this is, in part, because of the recent introduction of the virus to the South American population and associated large outbreaks. Based on the data presented here, we recommend that in endemic areas, CHIKV should be tested for in all patients presenting with acute neurological symptoms. Therefore, novel laboratory algorithms must be established to catch the real impact of arboviral infections.

## Figures and Tables

**Figure 1 viruses-14-00969-f001:**
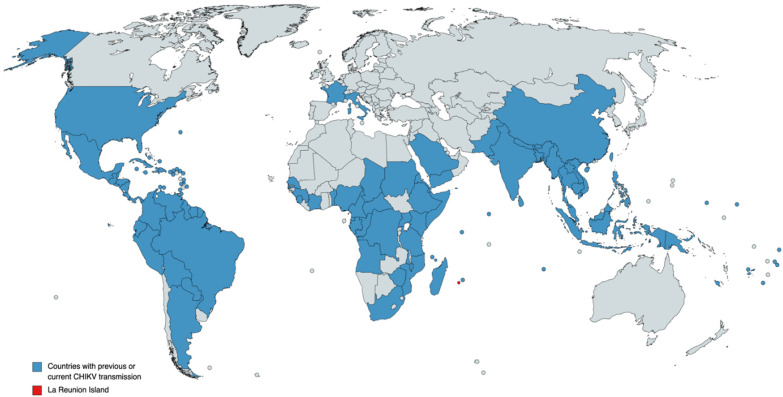
Countries with previous or current chikungunya transmission are blue-colored, according to the Centers for Disease Control and Prevention (CDC) (available at https://www.cdc.gov/chikungunya/geo/index.html, accessed on 28 April 2022), as reported by February 2022. La Réunion Island is red-colored, given its historical importance. Does not include imported cases. Although the United States territory of Alaska is blue-colored, no CHIKV transmission was documented in this area.

**Table 1 viruses-14-00969-t001:** Summary of all the current vaccines under a clinical trial evaluation, either Phase I/II or Phase III. These vaccines are discussed in more detail below.

Vaccine	Technology	Virus Strain	CHIKV Immunogen	Number of Doses	Development Stage	Developer	Reference
**VLA1553**	Live attenuated	LR2006 OPY1	Δ5nsP3	Single-dose	Phase III	Valneva Austria GmbH	NCT0483844 [110,111]
**BBV87**	Inactivated whole virus	ECSA genotype	Whole virus	Two doses	Phase II and Phase III	International Vaccine Institute/Bharat Biotech International	NCT04566484 and CTRI/2017/02/007755
**MV-CHIK-202**	Measles-vectored	pTM-MVSchw-CE3E26KE1	C-E3-E2- 6K-E1	One or Two doses	Phase II	Themis Bioscience	[112]
**VRC-CHKVLP059- 00-VP/PXVX0317**	VLP	West African strain/37997	E1, E2 and C	Two doses	Phase II and Phase III	US National Institutes of Health and PaxVax	NCT03483961, NCT02562482, and NCT05072080 and [113]
**CHIKV TSI-GSD-218**	Live attenuated	Southeast Asian strain/AF15561	Whole Virus	Single-dose	Phase II	US Army Medical Research Institute of Infectious Diseases and University of Maryland	[114,115]
**VAL-181388**	mRNA-based	-	mRNA encoding C, E3, E2, 6k, E1	Two doses	Phase I	ModernaTX, Inc	NCT03325075
**CHIKV001 (ChAdOx1-Chik)**	Adenoviral vector	Multiple	C, E3, E2, 6k, E1	Single-dose	Phase I	Jenner Institute, University of Oxford	[116]
**mRNA-1944**	mLNP-mRNA-based	-	mRNA encoding CHKV-24 IgG (monoclonal)	One or Two doses	Phase I	ModernaTX, Inc	[117]

Abbreviations: MV measles virus; VLP virus-like particle; Δ5nsP3 five viruses with deletions in the nsP3 protein region (Δ5nsP3 mutants); LNP lipid nanoparticle encapsulated.

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
