# Peer review of "A Review on Chikungunya Virus Epidemiology, Pathogenesis and Current Vaccine Development"

_viruses, 2022, doi:10.3390/v14050969_

Round 1

Reviewer 1 Report

Despite the already large number of reviews available on the subject, Vasconcelos de Lima Cavalcanti et al. present here an update of great interest to the scientific community with a focus on the progress of ongoing vaccine developments.

My comments and suggestions for improving the manuscript are as follows:

As the document does not indicate any numeroted lines, this work was made somewhat a little less easy.

  1. Introduction

Asymptomatic CHIKV infections do occur, but are rare and estimated at about 15% of the infected individuals.

Please provide one or more references to this assertion.

Acute symptomatic CHIKV disease resembles other common well-known ar-boviruses such as dengue virus (DENV) and Zika virus (ZIKV), which are frequently inaccurately diagnosed

From a semantic point of view, there is a confusion between the disease and the viruses involved. Change for example to  ‘arboviroses such as those from infections by dengue virus (DENV)…’

‘the initial virus infection’ and acute phase of disease!

CHIKV infection evolved to a more severe form of the disease

Care should be taken with this statement.

Did these severe forms observed during the Reunion and Indian Ocean epidemic reveal new capacities of a ‘new’ CHIKV or do they rather reflect a lack of data available at that time, notably on the description of clinical characteristics caused by this virus? 

As discussed below, the earlier epidemics were more modest, but above all had not been seriously described.

I suggest that you modify this sentence by reporting this lack of data with poorly described infectious mechanisms and associated pathologies, in the early 21st century.

775,000 population

Instead :

During this outbreak in 2006, it is estimated that CHIKV infected one third of the population of 775,000, causing 237 deaths.

This mutation, found in over 90% of viral sequences from Réunion Island, was associated with a slight increase in transmission by Ae. aegypti, showing an enhanced ability of the virus to replicate in insect cells that are naturally low in cholesterol.

This work on the effect of the mutation has rather shown an adaptation to Ae. albopictus. Especially as Ae. aegypti is almost absent from Réunion Island.

Thus, this mutation may have facilitated virus transmission, added to the fact that the virus was introduced in a region where the population was highly susceptible and without effective mosquito control measures.

Choose the word 'naive' rather than susceptible because the epidemic may have spread due to a lack of population immunity rather than a specific genetic background.

Then the wording without effective is inaccurate because vector control was deployed from the beginning of the epidemic. I would rather suggest :

the virus was introduced into an area where the population was naïve and where mosquito population control measures did not stop the outbreak.

Later in 2007- 2008 a CHIKV Asian genotype (closest to the ancient Asian genotype) appeared in Indonesia, this strain was identified in about 7% of the infection cases

It is worth mentioning that the Asian genotype has circulated quite a bit over the period with major outbreaks in Singapore, Taiwan… and Thailand in 2009.

In the paragraph on the emergence in the Americas from 2013, there is no longer any mention of modes of transmission. Can you recall whether the main vector was albopictus or aegypti?

Moreover, in the last paragraph on the risks, in particular on the co-circulations of arboviruses, it seems to me that a section on the expansion of vectors, towards temperate zones in particular, is missing. A discussion on the control of mosquito populations implemented in some countries would also be appropriate here.

  1. Pathogenesis

The first paragraph reviews the data on tropism and entry pathways of CHIKV into cells.

It is regretful to remain on this restrictive identification of ‘the’ entry receptor. I think it is important to point out that other entry pathways than the receptor-dependent endocytosis mentioned here, give CHIKV singularities for its tropism and for the resulting infection and pathogenesis.

You can add a lot to this section.

Thus the contribution of macropinocytosis is important for muscle tropism (doi: 10.1371/journal.pntd.0007610). Even more atypical: phagocytosis of apoptotic bodies by macrophages is the only way for CHIKV to target this type of cell niche, which is known to be important for the outcome of the infection and the onset of chronicity. (doi.org/10.1096/fj.10-164178).

CHIKV arthralgia chronification remains unclear

check the term chronification

Are the stated values in copies of RNA /mL or pfu/mL? And are there any differences in the viral loads observed during outbreaks with the ECSA virus versus the Asian lineage?

It seems that these values (106 in adults) are in a lower average range than that observed during the 2006 epidemics where up to almost 109 was frequently observed in viremic adults (for example reported in doi.org/10.1373/clinchem.2007.086595).

Perhaps this is worth highlighting and discussing? This could also correlate with the differences in the frequencies of developing severe chronic forms depending on the viral strains?

Regarding the next paragraph  and the assumptions of chronicity in relation to viral persistence and especially after the sentence ‘ These findings suggest that pathogenic CHIKV strains evade E2 domainB-neutralizing antibodies to establish persistence.’

It would be appropriate to mention this other hypothesis of a Trojan horse type strategy deployed by CHIKV, since it has been shown that this viral strategy results in a non-phlogistic infection of professional phagocytes, which would then become reservoirs for viral persistence...

doi.org/10.1096/fj.10-164178.

data about CHIKV virus persistence is virtually absent.

In any case, the data are patchy. However, much older work carried out on RRV provides a basis for drawing parallels between the two viruses, which are quite similar, including for chronic arthralgia.  For example, you could cite the work of Suhrbier doi: 10.1099/0022-1317-77-3-407 or doi: 10.1006/viro.2002.1587.

  1. Vaccines and treatment

A live attenuated vaccine based on the infectious clone of the La Reunion, strain CHIKVLR2006-OPY1, of a CHIKV African genotype was recently developed [90].

Check the reference : is it for the viral strain or for the ‘recent’ vaccine development?

In general, this paragraph is complete but lacks a small discussion of the advantages but perhaps disadvantages of using a vaccination whose objective is to obtain neutralizing antibodies. For example, the impact of ADE when antibodies become sub-neutralizing is not well known. A comment on this point would be welcome, as in the paper : doi: 10.1016/j.vaccine.2017.07.065

This discussion could be taken up in the concluding remarks.

Author Response

We thank all reviewers for their valuable contributions. Their comments helped to improve the quality of our manuscript. Below we are providing answers for their comments.

Major comments from reviewer #1

We accepted all the suggestions made by reviewer #1. We added a more detailed description regarding the Asian genotype epidemiology history in the introduction session. We corrected all the misused terms, as pointed by reviewer #1. In the pathogenesis section we added a more detailed description from cellular receptors, we changed the term "entry receptors" that was incorrectly used and we added some sentences describing macropinocytosis and engulfment of apoptotic bodies. We also added some discussion about viral loads from different strains and we provided information about viral titers as requested by this reviewer. All the remaining minor comments were also addressed. We are uploading a novel version of our manuscript where all the changes were yellow highlighted. We are grateful for this detailed review provided by reviewer #1.

Reviewer 2 Report

Senhora Vasconcelos de Lima Cavalcanti and colleagues summarize latest scientific updates in a review on Chikungunya virus epidemiology and pathogenesis and finally draw a picture of the most recent vaccine developments. The review will be a suitable introduction to the Special Issue.

This reviewer suggests that the authors uncouple the section on ChikV epidemics from the introduction with an own headline. Not too many readers are familiar with all the topographical locations mentioned in the manuscript. While most readers are able to spot the authors’ large home country, many may be unsure about the geographic location for instance of Gabon or the islands of the Lesser Antilles. Since the review does not come with a single figure, this reviewer suggests that the authors illustrate their article with a map with labels of all the relevant geographic entities. The chapter on ChikV pathogenesis may be slightly shortened in particular when elaborating on possible neurological consequences of a ChikV infection. The vaccine section of the manuscript is stronger, since it incorporates reflections of what we learned from the SARS-CoV-2 vaccines.

Minor comments:

The half sentence in brackets on page 6: “… c-type lectin receptor …”, which appears to having been copied and pasted, should be formatted to fit to the rest of the manuscript.

The term “Vaccinates” (page 10, line 8) should be replaced by the substantive “Vaccinees”.

Author Response

Major comments from reviewer #2

We appreciate the comments provided by reviewer #2. His suggestions contributed to improving the overall quality of our manuscript. We accepted the suggestion to include a figure and now we are uploading a novel version that includes a map, presenting the areas with reported chikungunya circulation. One of his suggestions was to make the pathogenesis section shorter, we apologize for this but we did the opposite, we accepted the suggestion made by the other reviewer and we included some new sentences about chikungunya entry and receptors, viral load, and pathogenesis comparisons among the different strains. Minor comments were all addressed. We are uploading a novel version of our manuscript where all the changes were yellow highlighted. We are grateful for this detailed review provided by reviewer #2.

Round 2

Reviewer 1 Report

The authors met all my expectations. The review is really complete and exciting.